# Grape Pomace as a Natural Source of Phenolic Compounds: Solvent Screening and Extraction Optimization

**DOI:** 10.3390/molecules28062715

**Published:** 2023-03-17

**Authors:** Rafaela P. Rodrigues, Ana M. Sousa, Licínio M. Gando-Ferreira, Margarida J. Quina

**Affiliations:** CIEPQPF, Department of Chemical Engineering, University of Coimbra, Rua Sílvio Lima, Pólo II—Pinhal de Marrocos, 3030-790 Coimbra, Portugal

**Keywords:** phenolic compounds, grape pomace, solid–liquid extraction, optimization, central composite design, sequential extraction

## Abstract

The optimization of extraction by using solvents of phenolic compounds (TPh) of grape pomace (GP) based on a central composite design was investigated. The GP was characterized, and preliminary assays were conducted with five different solvents (water, ethanol, acetone, methanol, and butanol) and the aqueous mixtures thereof. Ethanol and acetone were revealed to be the best solvents for TPh extraction. The main extraction parameters (temperature—T, time—t, solvent concentration, and liquid–solid ratio—L/S) were optimized by using a central composite design. The optimized conditions for the ethanol extraction (T = 60 °C, t = 1.5 h, L/S = 25 mL/gdryGP) and for acetone (T = 50 °C, t = 1.5 h, L/S = 25 mL/gdryGP) were determined. Single-stage extraction revealed a TPh of 45.18 ± 9.51 mgGAE/gdryGP for acetone and a TPh of 38.70 ± 3.64 mgGAE/gdryGP for ethanol. The characterization of the extracts revealed the presence of gallic acid, caffeic acid, syringic acid, vanillic acid, chlorogenic acid, and p-coumaric acid, where the concentration of the first three compounds stands out in all extracts. A three-stage extraction increased the yield of ethanol to 63.3 mg GAE/gdryGP and the yield of acetone to 59.2 mg GAE/gdryGP. Overall, both solvents allow the extraction of phenolic compounds of grape pomace, but ethanol is commonly considered a greener solvent for this purpose.

## 1. Introduction

The wine industry is a relevant economic sector not only for several European countries (the top three are Italy, Spain, and France) but also for other countries (e.g., United States, China, and Argentina). However, in addition to wine (desired product), a high amount of solid waste and wastewater are also produced [1,2,3,4].

Wine is an alcoholic beverage produced from a sequence of processes applied to grapes, including stalk removal, crushing, fermentation, and pressing [4,5]. Among the solid waste produced, grape pomace (GP), which is constituted by stalks, skins, and seeds, is one of the most relevant winery residues [2,4,5,6]. Indeed, GP has become an attractive natural source of phenolic compounds, which are well known for their high antioxidant activity and nutraceutical benefits [6,7]. Moreover, the natural phenolic compounds have revealed excellent properties for use as food preservatives, colorants, and antimicrobial activity [8,9,10], with valuable applications in other industries.

Bioactive compounds, such as phenolic molecules, can be recovered from a solid matrix through extraction processes assisted by solvents, microwaves, ultrasounds, etc. One of the most common and traditional extraction techniques for the recovery of phenolic compounds is solid−liquid extractions (SLE) [5,6,11]. SLE is based on mass transport phenomena that are affected mainly by the extraction solvent type, temperature, and extraction time [12,13]. The extraction time and temperature are among the most important parameters to be optimized to achieve high recoveries of compounds [5,6]. According to the published results, an increase in the operation time may enhance the extraction efficiency, since the release of phenolic compounds from the matrix occurs gradually [14,15]. However, an excessive contact time between solid and liquid can lead to the oxidation of phenolic compounds, which translates into a loss of their antioxidant activity [16]. In general, an increase in the extraction temperature also improves the phenolic extraction due to diffusion mass-transfer acceleration [12,14,15,17,18]. In fact, the extraction conditions can have a great impact on the yield and quality of the extracts produced [19]. According to the literature, solvent choice is one of the most critical operational parameters. Indeed, due to the polar nature of polyphenols, better extraction efficiencies can be obtained by using polar protic media such as hydroalcoholic solutions [20]. The most common solvents used for the extraction of phenolic compounds are methanol, ethanol, acetone, and water [19,20]. According to Yilmaz and Toledo (2006) [21], if pure solvents are considered, methanol exhibits the highest extraction capacity, followed by ethanol, acetone, and water. Several studies concluded that the addition of water as a co-solvent increases phenolic extraction when compared to pure solvents [15,18,22]. In fact, Yilmaz and Toledo (2006) [21] reported a total phenolic content that was three times higher when 50% acetone was used instead of pure acetone as a solvent. However, the published results are not irrefutable with regard to the ideal solvent for the extraction of polyphenols from winery residues, since the different combinations of the operational conditions can produce similar results. Additionally, there is a lack of studies that provide an optimization of the extraction process, and for that, the central composite design (CCD) can be a valuable tool. Data provided by a CCD can be analyzed through the desirability function to identify the optimized conditions [23]. This function is based on the transformation of the responses from different scales into a scale-free value (between 0 and 1). The value 0 is attributed when the factors give an undesirable response, while the value 1 corresponds to the optimized performance of the selected factors. This approach is an innovative aspect of the present work because by using the desirability function, the best condition for extracting phenolic compounds can be easily identified.

Grape pomace has become an attractive natural source of phenolic compounds through solvent extraction processes [5,24,25]. However, there is a lack of explorative and systematic approaches to selecting the best conditions for the extraction of these compounds.

In this context, this study aims to maximize the recovery of natural phenolic compounds from grape pomace via solid–liquid extraction after a screening of five different solvents and optimization of the desirability function obtained by using data from a central composite design. The main specific objectives are: (i) physical and chemical characterization of grape pomace; (ii) evaluation of the most effective solvents out of five to extract phenolic compounds; (iii) optimization of the extraction parameters; (iv) evaluation of the performance of the optimized extraction process regarding phenolic content, flavonoid content, antioxidant activity, and phenolic identification; (v) comparison between single-stage and three-stage extraction.

## 2. Results and Discussion

### 2.1. Characterization of Grape Pomace

The main physical and chemical characteristics of grape pomace are presented in Table 1. GP presents a pH of 3.88, which is comparable with the data commonly reported in the literature (pH 3.4–5.4) for this type of residue [26]. This acidic pH is a result of the presence of compounds such as phenolic acids or fatty acids. In general, a low pH is favorable to the conservation of bagasse until further utilization/valorization, as it limits the proliferation of microorganisms, such as fungi and bacteria, which happens favorably at a pH in the ranges of 4.5–5 and 6.5–7, respectively [27]. In the present study, the EC was 4.20 mS/cm, which is in agreement with the literature, in which EC values between 1.3 and 5.4 mS/cm have between reported [28,29]. EC corresponds to the ability of the aqueous solution to conduct electrical current and is directly related to the presence of dissolved solids solubilized from the waste in water [30].

Thus, the EC value indicates the greater or lesser tendency for solubilization of the solid matrix. TS and VS content of GP were 76.3 and 92.54% (on a dry basis) of the initial GP mass, respectively. These values agree with the literature, placing TS between 26.4 and 95.5% and VS in the range of 77.1 and 98.8% (dry basis) [31,32]. In addition to VS, COD also indicates the organic matter content of GP, correlating it with the amount of oxygen needed for its oxidation. The GP under study was characterized by 1397 mg O_2_/g GP (dry basis), which is similar to the data found in the literature, in which values between 1344 and 1406 mg O_2_/g GP (dry basis) have been indicated [33]. The value in question is high and reveals the potential negative impact on water resources or on the soil if not managed properly.

The measured fractions of lignin, cellulose, and hemicellulose are also in agreement with the literature. The expected ranges are total lignin content between 31.9 and 51.6%, cellulose in the range of 10.5 to 33%, and hemicellulose content between 6.1 and 21% [29,31,34,35,36]. The high lignin content is indicative of low biodegradability.

### 2.2. Preliminary Screening of Solvents

A screening of the solvents was performed to avoid an additional factor in the optimization phase. The solvents were selected based on data reported by other authors, who mention the use of ethanol, methanol, and acetone as the most suitable solvents for the solid–liquid extraction process [20]. In addition to these solvents, water, and butanol were also tested. The first was tested because it is the cheapest and “greenest” solvent. Although butanol is not widely referred to as a solvent for polyphenol recovery, it has been recognized as a low-environmental-risk solvent [37]. Aqueous mixtures (50%) of each organic solvent were also evaluated. The results for the total phenolic content extracted with each solvent are shown in Figure 1.

One of the most obvious conclusions that can be drawn is that aqueous solvent mixtures performed much better than pure solvents. It should be noted that, except for methanol, no statistical difference was observed when comparing pure solvents. In fact, several authors have referred to the synergistic effect of mixing an organic solvent with water for the recovery of polyphenols [20]. The solvent screening results revealed that the addition of 50% of water to each solvent increased the recovery of polyphenols between 2.3, in the case of methanol, and 12.5 times, in the case of acetone, when compared to the use of pure solvent. On the other hand, if the performance of solvents at a concentration of 50% with the extraction with water is compared, the addition of 50% solvent led to an increase in the recovery of phenolic compounds between 5.3, in the case of methanol, and 9.8 times, in the case of acetone.

Among the various solvents tested, acetone with a concentration of 50% showed the best performance, allowing the recovery of 22.5 mg GAE/g GP (dry basis), followed by a mixture of 50% ethanol, 50% butanol, and 50% methanol. Although butanol is a “green” solvent, its performance did not stand out in the recovery of phenolic compounds. Therefore, the mixtures of 50% aqueous ethanol and 50% aqueous acetone were selected for further optimization of the extraction process.

### 2.3. Grape Pomace Extraction Optimization

The optimization of the extraction process was based on a central composite design (CCD). In this case, the operating temperature (T), the contact time between solid and liquid phases (t), liquid–solid ratio (L/S), and solvent concentration in aqueous solution (Eth or Ac) were selected as factors, and the total phenolic content (TPh) was selected as a response variable, resulting in 52 experiments for each solvent (ethanol and acetone).

#### 2.3.1. Ethanol as Extraction Solvent

The total phenolic content was evaluated in all extracts that were obtained according to the CCD experiments with ethanol. Multiple regression analysis was used to analyze the data and obtain a mathematical relationship between the response variable (TPh) and the factors. The regression model includes main effects, second-order effects, and cross effects. Moreover, the statistically insignificant (*p* value > 0.05) effects were excluded from the model. The statistically significant main effects are represented in Figure 2a, while the comparison between experimental and predicted TPh is shown in Figure 2b.

The Pareto plot, Figure 2a, shows that the quadratic effect of ethanol concentration (Eth) is the most significant, followed immediately by the effect of temperature. These results demonstrate the very strong synergistic effect arising from the addition of water to a solvent [20]. The significant effect of temperature was also reported by Casagrande et al. [19], who found that the increase in this variable led to a higher amount of phenolic compounds extracted due to an enhanced diffusion rate. The effect of the L/S ratio was the fourth most significant. The cross-effect between the L/S ratio and the solvent concentration (EthxL/S) is also a key effect in phenolic compound extraction. The extraction time, on the other hand, is one of the factors with the least impact on the recovery of TPh.

Figure 2b shows the good predictive capacity of the multivariate model since the experimental data are close to the diagonal line. The regression model is considered statistically significant when the calculated *F*-value is at least three to five times greater than the *F*-critical [38]. In this study, since *F*-value = 516.85 and *F*-critical = 2.11, the former is 244.95 times higher than the *F*-critical, indicating the statistical significance of the obtained model (with *p* < 0.0001). The regression model to predict TPh (mg GAE/g dry GP) as a function of four experimental factors is described by Equation (2).
TPh = 32.50 + 5.03 *x*_1_ + 1.42 *x*_2_ + 2.29 *x*_3_ − 3.34 *x*_4_ − 1.44 *x*_1_ *x*_4_ − 1.75 *x*_3_ *x*_4_ + 3.91 *x*_1_^2^ − 3.65 *x*_3_^2^ − 16.07 *x*_4_^2^(1)
where *x*_1_ = (T − 45)/15; *x*_2_ = (t − 1)/0.5; *x*_3_ = (L/S − 20)/10; and *x*_4_ = (Eth − 55)/35.

The response surface methodology allows for the evaluation of the relationship between the different pairs of the model variables. In this present study, through the CCD design, the effects of the L/S ratio and Eth, as shown in Figure 3a, and the joint effect of T and Eth, as shown in Figure 3b, were evaluated.

The zone of high TPh extraction was positively correlated to temperature, and an intermediate L/S seems favorable. Indeed, a pronounced curvature is observed for ethanol concentration. The results demonstrate that the central Eth (%) enhances the extraction of the phenolic compounds.

Figure 4 depicts the behavior of each extraction factor individually and the values that allow the maximum TPh recovery.

For both Eth and L/S, the conditions that optimize the extraction of phenolic compounds are well evident, since the curves show a defined maximum. Moreover, the extraction time was found to be a less relevant variable since only a small variation (2.8 mg GAE/g dry GP) in the response variable (TPh) is observed when time increases from 0.5 to 1.5 h. According to Amdoun et al. [23], the desirability function has been one of the most used approaches for factor optimization. This function is based on the transformation of the responses from different scales into a scale-free value (between 0 and 1). The value 0 is attributed when a factor gives an undesirable response, while the value 1 corresponds to the optimized performance of the selected factors. Based on the maximization of the desirability function (Figure 4e), it is possible to obtain the optimized conditions for the extraction process with ethanol-based mixtures: a temperature of 60 °C, an extraction time of 1.5 h, 49.1% ethanol, and L/S of 23.6 mL/g dry GP, which resulted in a predicted TPh of 43.7 mg GAE/g dry GP.

Da Porto and Natolino [39] reported that for a room-temperature extraction (22 °C), an optimized L/S 50 mL/g, an extraction time of 22 h, and an ethanol concentration between 50 and 60% led to a TPh of about 24 mg GAE/g (dry basis). Moreover, Kwiatkowski et al. [40] reported L/S 6.6 mL/g, extraction time of 1 h, and 60% of ethanol as the optimized extraction conditions for achieving 10.23 mg GAE/g dry GP.

#### 2.3.2. Acetone as Extraction Solvent

Furthermore, the extraction of the phenolic compounds with acetone was performed. The TPh obtained according to CCD experiments was evaluated in all extracts, and the results were interpreted through a multiple regression model. The statistically insignificant (*p* value > 0.05) effects were excluded from the model. The statistically significant main effects are represented in Figure 5a, and the comparison between the experimental and the predicted TPh is shown in Figure 5b.

Similar to the ethanol regression model, the quadratic effect of solvent concentration was also the most dominant in the acetone extraction, which is consistent with the results of the preliminary experiments. In the case of acetone mixtures, the L/S and the contact time prevail over the temperature effect.

Figure 5b shows the high predictive capacity of the model, for which a good agreement between the experimental and predicted TPh is also observed for the acetone extraction. Indeed, the regression model can be considered statistically significant since *F*-value= 702.46, which is 344.36 times greater than the *F*-critical (2.04, at 95% confidence level), and the model *p* value obtained was <0.0001. The regression model that expresses the TPh (mg GAE/g dry GP) as a function of the four experimental factors is described by Equation (3).
TPh = 46.22 + 1.16 *x*_5_ + 2.43 *x*_6_ + 2.50 *x*_7_ − 4.27 *x*_8_ + 1.17 *x*_5_ *x*_6_ + 1.17 *x*_5_ *x*_7_ − 1.325 *x*_5_ *x*_8_ + 0.88 *x*_6_ *x*_7_ + 3.63 *x*_5_^2^ − 6.52 *x*_7_^2^ − 21.73 *x*_8_^2^(2)
where *x*_5_ = (T − 40)/10; *x*_6_ = (t − 1)/0.5; *x*_7_ = (L/S − 20)/10; and *x*_8_ = (Ac − 55)/35.

This model is more complex than the one obtained with ethanol since it has more cross-effects. The effects on TPh of the liquid–solid ratio (L/S) and acetone concentration (Ac), Figure 6a, and the joint effect of temperature (T) and Ac (%), Figure 6b, were evaluated through a response surface methodology.

The optimized zone is observed for a high extraction temperature, an intermediate L/S, and an acetone concentration that allows for a higher TPh.

Figure 7 depicts the behavior of each extraction factor individually and the maximum TPh extraction. Based on the maximization of the desirability function, the optimized conditions with acetone-based mixtures were identified, corresponding to a temperature of 50 °C, an extraction time of 1.5 h, 50.5% acetone, and an L/S of 23.6 mL/g dry GP, which resulted in a predicted TPh of 55.8 mg GAE/g dry GP.

### 2.4. Validation of the Models

After the optimization of the process, both ethanol and acetone models were validated by performing five untested experimental conditions, including near-optimized point conditions. Table 2 summarizes the experimental conditions, as well as the obtained and predicted results.

The experimental and predicted data of the validation set points can be observed in Figure 8.

It is possible to observe from Figure 8 that both models can predict the phenolic content extracted under different conditions from those within the model construction.

### 2.5. Grape Pomace Extraction Performance

Besides phenolic content (TPh), total flavonoid content (TFC) and antioxidant activity (IC_50_) were also evaluated in triplicate under the optimized conditions (ethanol: T = 60 °C, L/S = 25, t = 1.5 h, and concentration of 50%; acetone: T = 50 °C, L/S = 25, t = 1.5 h, and concentration of 50%). The results are shown in Table 3.

The optimized extraction conditions are similar for the two solvents, except for temperature. Despite using a temperature 10 °C lower for acetone than in the extraction with ethanol, the acetone allowed for the extraction of 45.18 ± 9.51 mg GAE/g dry GP and ethanol 38.70 ± 3.64 mg GAE/g dry GP. However, according to the Tuckey HSD test, the difference in the results from the extraction with ethanol and acetone is not statistically significant. The same is true for TFC and IC_50_. Casagrande et al. [19] identified acetone as the best solvent when compared to others. Moreover, acetone extraction produced extracts with a higher concentration of flavonoids and with higher antioxidant activity (lower IC_50_).

Six phenolic compounds were identified and quantified in GP extracts for both solvents (50% ethanol and 50% acetone) via liquid chromatography (Table 4). Namely, it was possible to identify the presence of gallic acid, chlorogenic acid, vanillic acid, caffeic acid, syringic acid, and p-coumaric acid. Among the phenolic acids identified in the GP extracts by using 50% ethanol as the solvent, the gallic acid (22.83 ± 0.23 mg/100 dry g GP), followed by the caffeic acid (20.00 ± 0.09 mg/100 dry g GP), the syringic acid (16.36 ± 0.03 mg/100 dry g GP), the vanillic acid (5.98 ± 0.06 mg/100 dry g GP), the chlorogenic acid (2.06 ± 0.11 mg/100 dry g GP), and p-coumaric acid (1.67 ± 0.13 mg/100 dry g GP) stand out.

Moreover, the GP extracts using 50% acetone as the solvent revealed a similar composition, except for caffeic acid, which presented higher concentrations (38.22 ± 0.25 mg/100 dry g GP). Similar compositions of grape pomace extracts were found in the literature [19].

Additionally, the efficiency of both solvents in the optimized conditions was determined over three consecutive extractions. The TPh and TFC contents obtained are depicted in Figure 9.

The results demonstrated that the highest content of phenolic compounds was extracted in the first stage. In the case of acetone, approximately 76% of TPh and TFC are recovered in the first stage, followed by 17% in the second and 7% in the last stage. A similar pattern is observed for ethanol extractions (65%, followed by 22% and 13%). Kwiatkowski et al. [40] obtained analogous results over three consecutive ethanolic extractions. The global results of the consecutive extractions led to a TPh yield of 63.3 mg GAE/g dry GP with ethanol and 59.2 mg GAE/g dry GP with acetone. These results suggest that both solvents are suitable to extract phenolic compounds from grape pomace from a technical point of view. Other factors, such as cost, recovery of solvent, and environmental impact, may be taken into account to select the best solvent.

Adeyi et al. [41] studied a solvent extraction of *Carica papaya* L. leaves (CPL) by using water as the solvent (L/S of 40.25 mL/g) at 35 °C for 100 min. According to the results, it was possible to obtain an extract with a phenolic content of 74.65 mg GAE/g dry CPL with a yield of 18.76% (*w*/*w*). Furthermore, the uncertainty and sensitivity of the process and cost parameters on the total unit production cost (UPC) regarding the phenolic extraction of CPL were evaluated. The authors concluded that, among all the parameters evaluated, the extraction time, solid-to-liquid ratio, and extraction temperature exhibited significant impacts on the unit production cost. The increase in these parameters caused an increase in the UPC. In fact, the variance observed in the total cost was mainly due to the extraction time (47.1%), L/S ratio (42.8%), and extraction temperature (0.5%). According to the above stated, without considering the solvent cost, the extraction process cost for the two solvents would be similar since the difference between them is simply the temperature (10 °C), which is the parameter with the lowest impact on the UPC. Moreover, the authors estimated the total cost of the production of dried phenolic-rich bioactive extracts [41]. The analysis was performed while taking into account the Nigerian context and a production scale between 638 and 20,431 kg of dry extracts per year. The results indicated that the cost may reach about 525 USD/kg extract for the most economically feasible solution, which corresponds to a plant capacity of 19,857 kg of dry extracts per year.

## 3. Material and Methods

### 3.1. Materials

The grape pomace (GP) sample tested in the laboratory was collected from a red wine producer from the central region of Portugal after the fermentation of three Portuguese grape varieties (Trincadeira, Baga, and Alfrocheiro). The GP sample (ca. 10 kg) was dried at 40 °C until it reached a constant weight, milled (Retsch, model 5657, Germany) until particle size < 1 mm, and stored in a sealed and dry environment until further utilization. The solvents tested were ultrapure water (Direct-Pure Water System, Interlab, China), methanol (Carlo Erba Reagents, 99.9%, France), ethanol (Honeywell, 99.8%, Germany), acetone (Fisher Chemical, 99.8%, Belgium), and butanol ((Fisher Chemical, 99.5%).

### 3.2. Extraction Experiments and Preliminary Screening of Solvents

The extraction experiments on the GP were performed by using 80 mL closed vessels, in a thermostatic shaking water bath (VWR, 18 L Shaking Bath, United Kingdom) that was agitated at 160 rpm. After each extraction experiment, the suspension was immediately filtered through a 0.45 µm membrane filter, and both the extract and the solid fraction were dried at 40 °C in an oven and stored for further analysis.

Before the extraction optimization tests, preliminary tests were carried out to select the most appropriate solvent. Five solvents were initially tested, namely, water, ethanol, acetone, methanol, and butanol, in different concentrations (pure and diluted to 50% *v*/*v*). These experiments were performed in triplicate in pre-set values of time (t): 1 h, temperature (T): 50 °C, and liquid–solid ratio (L/S): 10 mL/g dry GP and shaken in a water bath at 160 rpm based on the different conditions reported in the literature [5].

### 3.3. Optimization of Grape Pomace Extraction

The optimization of the extraction process involved a central composite design (CCD) to evaluate the influence of four factors on the recovery of phenolic compounds, namely, the temperature, time, L/S ratio, and solvent concentration. This type of experiment design enables obtaining a surface response and a predictive second-order effects model, according to Equation (1):(3)$$Y=\beta_{0}+\overset{4}{\underset{i=1}{\sum }}\beta_{i}x_{i}+\overset{4}{\underset{i=1}{\sum }}\overset{4}{\underset{j=1}{\sum }}\beta_{ij}x_{i}x_{j}+\overset{4}{\underset{i=1}{\sum }}\gamma_{i}x_{i}{}^{2}$$
where *Y* is the response variable, which, in this study, is referred to as total phenolic content (TPh); *x_i_* and *x_j_* represent the factors selected for optimization; *β*_0_ is the independent term; and *β_i_*, *β_ij_*, and *γ_i_* are the coefficients of the model variables.

Temperature restrictions were considered based on solvent characteristics and safety measures. The optimization tests were performed in duplicate, and the experimental conditions are shown in Table 5.

### 3.4. Analytical Methods

Total solids (TS) and the organic matter quantified as volatile solids (VS) were determined according to APHA (1992) Standard Methods [42]. Briefly, TS was quantified via GP at 40 °C until a constant weight was reached, and VS was evaluated by calcinating the sample in a furnace at 550 °C for 2 h. The natural pH and the electrical conductivity (EC) of the residues were measured (Consort C1020) in a suspension at an S/L ratio of 1:10 (kg:L). Total chemical oxygen demand (COD) was determined via the close reflux method according to APHA (1992), using potassium dichromate as an oxidant. Elemental composition (CNHS) was assessed through an Elemental Analyzer NA 2500 equipment, and the oxygen concentration was calculated by the difference between VS content and the sum of CNHS.

Total Kjeldahl Nitrogen (TKN) was determined by taking 0.5 g of GP followed by digestion (DKL Fully Automatic Digestion Unit from VELP Scientifica, Italy), distillation (UDK Distillation Unit from VELP Scientifica), and titration.

Lignin, cellulose, and hemicellulose content were obtained in triplicate according to the National Renewable Energy Laboratory (NREL) procedure [43]. In short, 3 mL of 72% of sulfuric acid was added to 300 mg of the sample, and the suspension was incubated in a bath at 30 °C for 1 h, with stirring every 5 to 10 min. Then, 84 mL of deionized water was added to the previous suspension, diluting the acid to 4%. The diluted suspension was autoclaved at 121 °C for 1 h and filtrated. Acid-insoluble lignin content was obtained by weighing the remaining solids that were dried at 105 °C. Acid-soluble lignin was measured by using an aliquot of the filtrate via a UV-Vis spectrophotometer at 205 nm (PG Instruments T60, United Kingdom). Structural carbohydrates were analyzed via high-performance liquid chromatography (HPLC-RI, Knauer model K-301, Germany), as described elsewhere [44]. Cellulose and hemicellulose contents were calculated based on the degradation products resulting from hydrolysis.

Total phenolic content (TPh), which was expressed in mg of gallic acid equivalents (GAE) per g of dry GP, was measured in triplicate by using a modified colorimetric Folin and Ciocalteu method in which 20 µL of the re-suspended extract was mixed with 1580 µL of distilled water, 100 µL of Folin–Ciocalteu’s reactant, and 300 µL of Na_2_CO_3_ solution. The absorbance was measured via a UV/vis spectrophotometer (PG Instruments T60) at 765 nm after 30 min of reaction at 40 °C.

Total flavonoid content (TFC) content, which was expressed in mg of catechin equivalents (Cat) per g of dry GP, was determined, in triplicate, by mixing 20 µL of re-suspended extract, 1460 µL of distilled water, 60 µL of NaNO_2_ (5%), 60 µL of AlCl_3_ (10%), and 400 µL NaOH (1 M). The absorbance of the mixtures was measured at 510 nm by using a T60 UV-Vis spectrophotometer.

The radical scavenging assay with 2,2-diphenyl-1-picryhydrazyl (DPPH) was carried out, in triplicate, for evaluating the antioxidant potential of the extracts [45]. Samples were diluted in ethanol, and antioxidant activities were expressed as IC_50_ (mg/mL), which is defined as the extract concentration able to scavenge 50% of the DPPH radical, which is also designated as reducing capacity.

The identification and quantification of the phenolic compounds present in the extracts were performed in an HPLC system (Waters separation model 2695 with a Waters 2487 dual-absorbance detector) by using a Brisa LC2 C18 column (250 × 4.6 mm id, 5 μm, Spain). The chromatographic conditions were adapted from those described elsewhere [46]. The mobile phase (A) is water-adjusted with 1.17 mL/L of phosphoric acid and acetonitrile (B) operated at a flow rate of 1 mL/min, with an elution gradient applied for a duration of 42 min as follows: 100–89% A (0–3 min), 89–89% A (3–26 min), 89–80% A (26–39 min), and 80–100% A (39–100 min). The column temperature was set to 30 °C, and the sampler was set at 25 °C. All dried extracts were solubilized in water at concentrations between 10 and 15 gdb/L. Samples were filtered by a 0.45 μm microfilter, and the injection volume was 30 μL. The chromatographic profiles were measured at 215 and 280 nm. To identify the retention times of the phenolic compounds, standard solutions of several phenolic compounds commonly present in winery extracts were used: gallic acid (Alfa Aesar, 98%, Germany), chlorogenic acid (Sigma-Aldrich, >99%, Germany), vanillic acid (TCI Chemicals, >98%, Belgium), caffeic acid (TCI Chemicals, >98%), syringic acid (Thermo Scientific Chemicals, 97%, Germany), p-coumaric acid (Sigma-Aldrich, 99%), and ferulic acid (Sigma-Aldrich, 98%).

### 3.5. Statistical Analysis

The design of the experiments was analyzed by using the JMP^®^ Pro 15, and a significance level of 95% was adopted to determine the relevant model factors. The fitting capacity of the models was evaluated through the use of analysis of variance (ANOVA).

A Tuckey HSD test with a confidence level of 95% was employed to analyze the significant differences in the total phenolic content that was obtained in the extraction with different solvents and the performance of three consecutive extractions.

## 4. Conclusions

Aiming at the extraction of phenolic compounds (TPh) from grape pomace, five solvents (water, methanol, ethanol, butanol, and acetone), which were tested in their pure and diluted forms, were screened in preliminary experiments. The dilution of the solvents led to higher recovery of phenolics when compared to pure solvents. Globally, acetone (50% *v*/*v*) and ethanol (50% *v*/*v*) were found to be the best solvents. The effect of solvent concentration, liquid–solid ratio (L/S), temperature (T), and time (t) was optimized to maximize the TPh extraction through a central composite design (CCD) for both acetone and ethanol. The multiple regression models obtained contain non-linear terms, and the fitting to data was considered adequate for further predictions. The optimized extraction conditions for each solvent were identified by using the desirability function. For both solvents, the optimized conditions are 1.5 h, 50% solvent concentration, and 25 mL/g dry GP. The optimized temperatures of 60 and 50 °C were found for ethanol and acetone, respectively. Single-stage extraction in the optimized conditions revealed 45.18 ± 9.51 mg GAE/g dry GP for acetone and 38.70 ± 3.64 mg GAE/g dry GP for ethanol. Among the phenolic compounds, gallic acid, caffeic acid, and syringic acid stand out for both solvents. By using a three-stage extraction, 63.3 and 59.2 mg GAE/g dry GP for ethanol and acetone were achieved. Overall, both solvents revealed a good performance for the extraction of phenolic compounds from grape pomace.

## Figures and Tables

**Figure 1 molecules-28-02715-f001:**
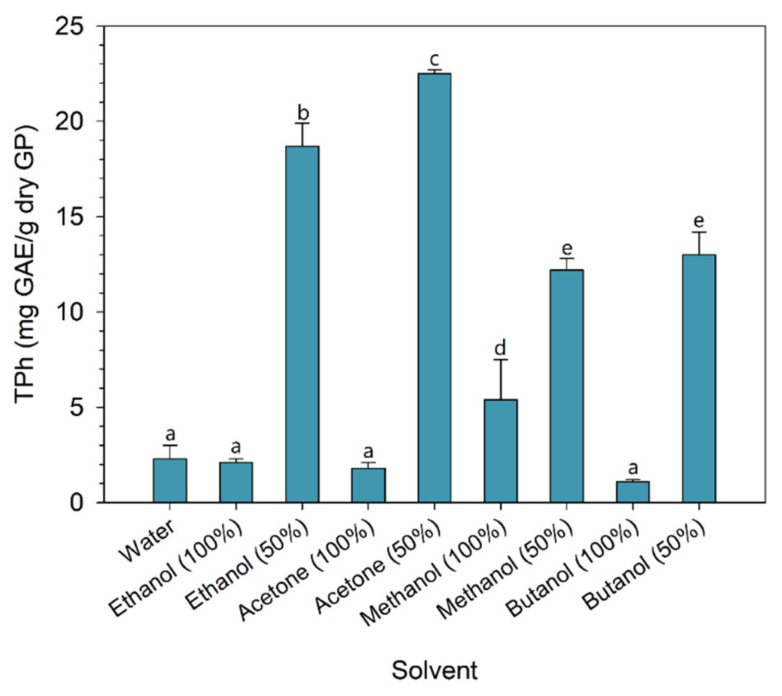
Total phenolic content (mean ± standard deviation) extracted for different solvents (statistically significant differences are marked with different lowercase letters, at the *p* value < 0.05).

**Figure 2 molecules-28-02715-f002:**
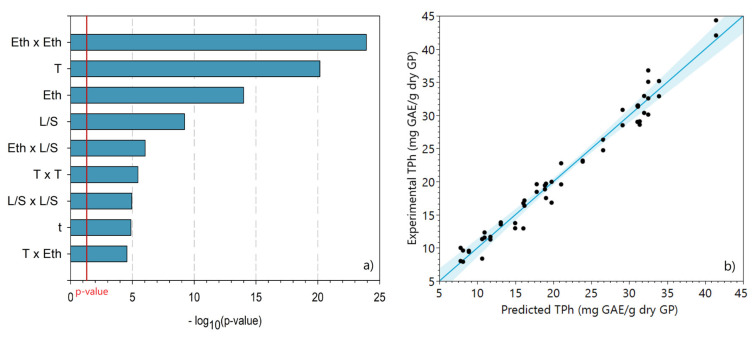
(**a**) Pareto plot of the statistically significant effects of the variables and their interactions for ethanol extraction. (**b**) Comparison between experimental and predicted total phenolic content obtained from the central composite design of ethanol.

**Figure 3 molecules-28-02715-f003:**
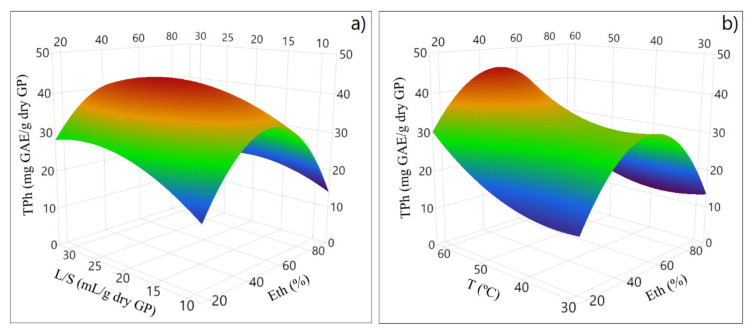
Response surface plots for the effect of (**a**) liquid–solid ratio (L/S) and ethanol concentration (Eth); and of (**b**) temperature (T) and ethanol concentration (Eth) on total phenolic content (TPh).

**Figure 4 molecules-28-02715-f004:**
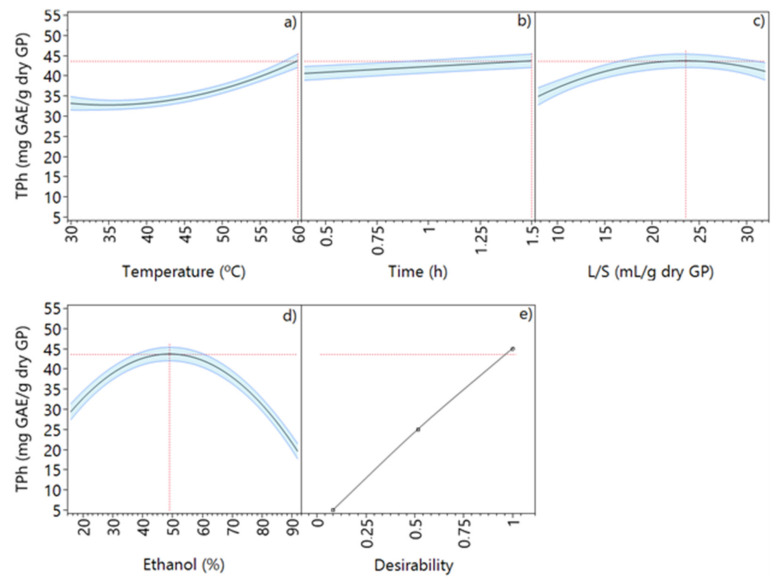
Total phenolic content prediction profiler of ethanol extraction at optimized conditions: (**a**) temperature, (**b**) time, (**c**) liquid–solid ratio, (**d**) ethanol concentration, and (**e**) maximized desirability.

**Figure 5 molecules-28-02715-f005:**
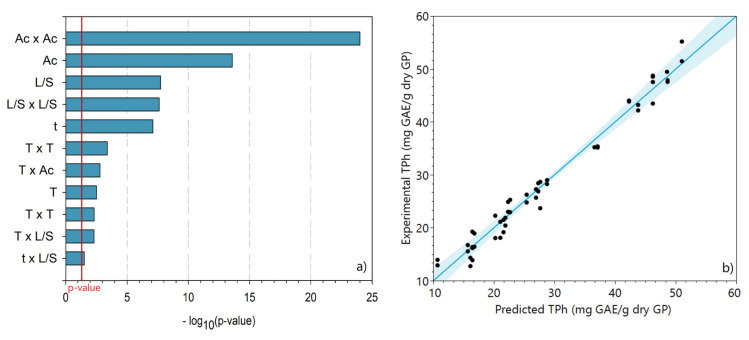
(**a**) Pareto plot of the statistically significant effects of the variables and their interactions for acetone extraction. (**b**) Comparison between experimental and predicted total phenolic content obtained from the central composite design of acetone extraction.

**Figure 6 molecules-28-02715-f006:**
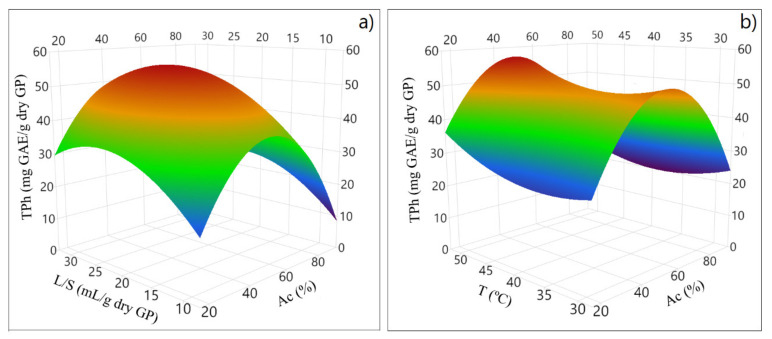
Response surface plots for the effect of (**a**) liquid–solid ratio (L/S) and acetone concentration (Ac) and of (**b**) temperature (T) and acetone concentration (Ac) on total phenolic content (TPh).

**Figure 7 molecules-28-02715-f007:**
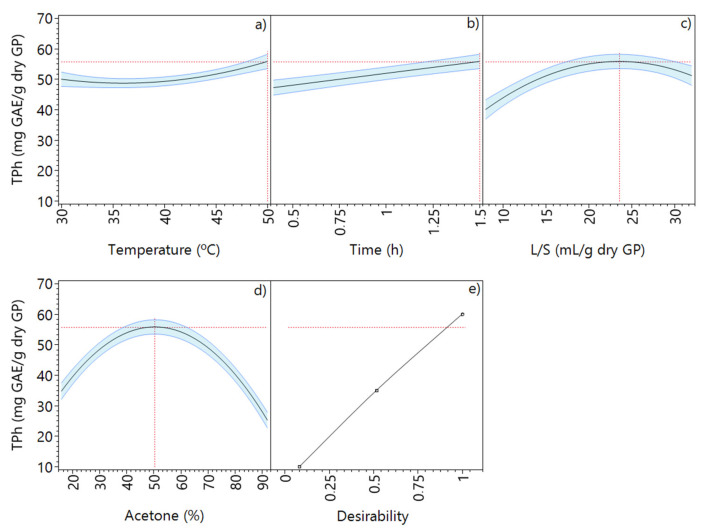
Total phenolic content prediction profiler of acetone extraction at optimized conditions: (**a**) temperature, (**b**) time, (**c**) liquid–solid ratio, (**d**) ethanol concentration, and (**e**) maximized desirability.

**Figure 8 molecules-28-02715-f008:**
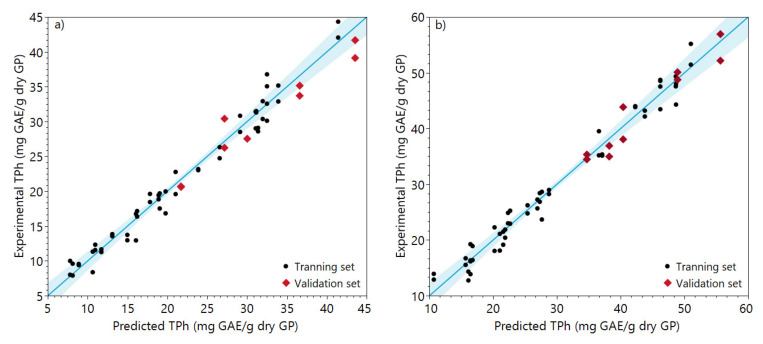
Comparison between experimental and predicted total phenolic content obtained from the central composite design of (**a**) ethanol and (**b**) acetone extraction.

**Figure 9 molecules-28-02715-f009:**
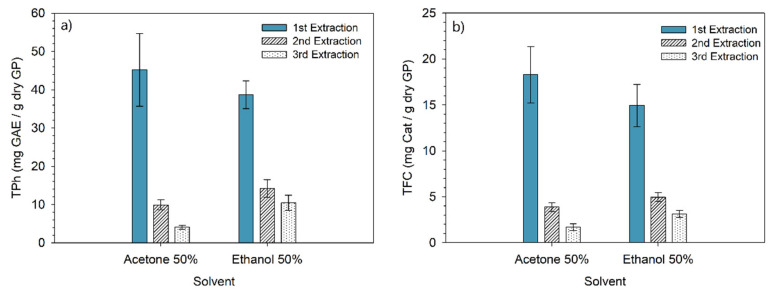
(**a**) Total phenolic content (mean ± standard deviation) and (**b**) total flavonoid content (mean ± standard deviation) of grape pomace obtained over three consecutive extractions with ethanol and acetone.

**Table 1 molecules-28-02715-t001:** Properties of grape pomace tested in the present study.

Parameter	Value	Parameter	Value
pH	3.88 ± 0.02	TS (%)	76.30 ± 5.30
EC (mS/cm)	4.20 ± 0.05	VS (%) ^a^	92.51 ± 0.06
COD (mg O_2_/g) ^a^	1397 ± 50	Lignin (%) ^a^	50.62 ± 1.40
C (%) ^a^	48.04 ± 0.01	Cellulose (%) ^a^	19.04 ± 0.31
N (%) ^a^	1.42 ± 0.01	Hemicellulose (%) ^a^	8.18 ± 0.02
O (%) ^a^	34.60 ± 0.03	TKN (mgN/g) ^a^	22.51 ± 2.13
H (%) ^a^	6.15 ± 0.02	Empirical formula	C_39_._3_H_60_._5_O_21_._2_N

^a^ in dry basis (db).

**Table 2 molecules-28-02715-t002:** Levels selected for each factor for the validation assays of ethanol and acetone extraction models and respective experimental (mean ± standard deviation) and predicted responses.

T (°C)	t (h)	L/S (mL/g Dry GP)	Solvent (%)	TPh (mg GAE/g Dry GP)	
				Experimental	Predicted
Ethanol extraction					
60	1.50	25	50	40.4 ± 1.8	43. 6
50	1.50	25	50	34.4 ± 1.0	36.6
50	1.25	15	30	28.3 ± 2.9	27.2
40	0.75	15	30	20.6 ± 0.1	21.7
50	1.00	20	70	27.5 ± 0.01	30.0
Acetone extraction					
50	1.50	25	50	54. 6 ± 3.4	55.6
40	1.50	25	50	49.4 ± 0.9	48.9
45	1.25	15	30	35.9 ± 1.4	38.2
35	0.75	15	30	34.9 ± 0.6	34.7
40	1.00	20	70	40.9 ± 4.1	40.4

T—temperature; t—time; L/S—liquid–solid ratio; TPh—total phenolic content; GAE—gallic acid equivalents; GP—grape pomace.

**Table 3 molecules-28-02715-t003:** Total phenolic content, total flavonoid content, and reducing capacity (mean ± standard deviation) of grape pomace ethanol and acetone extraction at optimized conditions.

Solvent	TPh (mg GAE/g Dry GP)	TFC (mg Cat/g Dry GP)	IC_50_ (mg/mL)
50% Ethanol	38.70 ± 3.64 ^a^	14.94 ± 2.29 ^b^	22.25 ± 3.00 ^c^
50% Acetone	45.18 ± 9.51 ^a^	18.29 ± 3.07 ^b^	14.93 ± 3.81 ^c^

TPh—total phenol content; TFC—total flavonoid content; IC_50_—reducing capacity. Statistically significant differences in lowercase letters at the *p* value < 0.05.

**Table 4 molecules-28-02715-t004:** Average and standard deviation (n = 3) of the concentrations of the phenolic compounds determined via HPLC in grape pomace extracts.

Solvent	Compounds	Retention Time (min)	Concentration (mg/100 Dry g GP)
50% Ethanol	Gallic acid	6.767	22.83 ± 0.23
	Chlorogenic acid	12.193	2.06 ± 0.11
	Vanillic acid	14.302	5.98 ± 0.06
	Caffeic acid	14.762	20.00 ± 0.09
	Syringic acid	16.113	16.36 ± 0.03
	p-Coumaric acid	25.568	1.67 ± 0.13
50% Acetone	Gallic acid	6.869	25.01 ± 0.19
	Chlorogenic acid	12.230	2.75 ± 0.09
	Vanillic acid	14.358	6.42 ± 0.04
	Caffeic acid	14.789	38.22 ± 0.25
	Syringic acid	16.140	16.18 ± 0.04
	p-Coumaric acid	25.570	1.03 ± 0.12

**Table 5 molecules-28-02715-t005:** Experimental values and coded levels of the independent variables tested in the central composite design.

Factors	Code Units	Coded Level		
		−1	0	1
Ethanol extraction				
Temperature (°C)	T	30	45	60
Time (h)	t	0.5	1.0	1.5
Liquid–solid ratio (mL/g dry GP)	L/S	10	20	30
Ethanol concentration (% *v*/*v*)	Eth	20	55	90
Acetone extraction				
Temperature (°C)	T	30	40	50
Time (h)	t	0.5	1.0	1.5
Liquid–solid ratio (mL/g dry GP)	L/S	10	20	30
Acetone concentration (% *v*/*v*)	Ac	20	55	90

## Data Availability

Not applicable.

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
