# Peer review of "Grape Pomace as a Natural Source of Phenolic Compounds: Solvent Screening and Extraction Optimization"

_molecules, 2023, doi:10.3390/molecules28062715_

Round 1

Reviewer 1 Report

1.     Line 20, the statement “ Overall, both solvents allowed the”- statement means allows? Rewrite the word choice

2.     Line 61-64, “However, the ……can be a valuable tool”- authors stating that “published results are not irrefutable 61 about the ideal solvent for the extraction of polyphenols”- Did authors having any justification for statement. There are many articles might have discussed the same?

 3.     What is the novelty or significance of this work?

 4.     Line 1487, Gaspar et al. (2019) [28], line 170 - Mulinacci  et al. (2001) -use the citation as per journal style. Applicable to other places as well where the similar issues exist

 5.     Line 154, Na2CO3 solution- check the typeset. Similarly line 159, NaNO2 (5%), 60 μL of AlCl3 (10%)- check type set

Reviewer 2 Report

Introduction

Please, remove the sentence “Grape pomace (stalks, skins, and seeds) is one of the most relevant winery residues 73 and has become an attractive natural source of phenolic compounds. This can be recov- 74 ered from a solid matrix through extraction processes. Other studies explore the extrac- 75 tion of phenolic compounds from grape pomace using solvents [5, 24, 25]. However, there 76 is a lack of explorative and systematic approaches to selecting the best conditions for the 77 extraction of these compounds” since it is redundant.

Materials

Please, specify the type of red grape and the reason of that choice.

Please, specify if a previous study on the optimization of drying temperature has been performed and the reason of the choice of 40°C.

Authors say that “and both the extract and the solid fraction were dried at 40°C in an oven and 101 stored for further analysis.” Are they sure that it is the best method to preserve the phenolic content?

Reviewer 3 Report

This manuscript presents a study focused on the use of grape pomace as a source of phenolic compounds. The extraction parameters were studied using an experimental design approach, which is undeniably the best way to evaluate the influence of variables within a certain process. 

The study design is generally well constructed. Still, the use of total polyphenol content for extraction evaluation significantly limits the impact of this study, since there is no quantitative information of the effect of extraction parameters on the 6 polyphenols that were quantified by HPLC. It would have been much more interesting to see these effects measured by HPLC throughout the whole experiment.

There are still some issues which need adjusting before publication:

1. The aim of this study is somehow unclear. The focus is put on the extraction of phenolic compounds, yet there is no clear goal describing why this study was carried.

2. The first paragraph of the introduction should be placed just before the paragraph starting at line 73.

3. The use of the terms "optimal" and "optimum" should be avoided when talking about results determined in certain experimental conditions. These terms represent an ideal outcome. Instead, the term "optimized" should be used, since it better reflects what was done.

4. Chapter 2. Materials and methods. There are some details which are missing, such as:

- the purity of the water that was used and how it was obtained;

- details regarding the analytical standards used for HPLC measurements;

- the producers of all instruments, chemicals and reagents should be mentioned.

5. other minor issues:

- Line 154 and elsewhere: subscript when needed (e.g. Na2CO3 not Na3CO3)

- Table 2. all measured values should be presented with the same number of decimals

- Figure 1: what is the meaning of the lowercase letters above each column?

- Line 381: term "ethanol" should be replaced with "acetone", since its the discussion regarding the extraction optimization using acetone as extraction solvent

- Line 460: plant names should be written with italic font
